# NPFF Decreases Activity of Human Arcuate NPY Neurons: A Study in Embryonic-Stem-Cell-Derived Model

**DOI:** 10.3390/ijms23063260

**Published:** 2022-03-17

**Authors:** Lola Torz, Kristoffer Niss, Sofia Lundh, Jens C. Rekling, Carlos Damian Quintana, Signe Emilie Dannulat Frazier, Aaron J. Mercer, Anda Cornea, Charlotte Vinther Bertelsen, Marina Kjærgaard Gerstenberg, Ann Maria Kruse Hansen, Mette Guldbrandt, Jens Lykkesfeldt, Linu Mary John, J. Carlos Villaescusa, Natalia Petersen

**Affiliations:** 1In Vitro Obeisty Research, Novo Nordisk, 2760 Måløv, Denmark; amkh@novonordisk.com (A.M.K.H.); mgul@novonordisk.com (M.G.); napt@novonordisk.com (N.P.); 2Section for Experimental Animal Models, Department of Veterinary and Animal Sciences, University of Copenhagen, 1870 Frederiksberg, Denmark; jopl@sund.ku.dk; 3Bioinformatics & Data Mining, Novo Nordisk, 2760 Måløv, Denmark; kfni@novonordisk.com; 4Pathology & Imaging, Novo Nordisk, 2760 Måløv, Denmark; qsln@novonordisk.com; 5Department of Neuroscience, University of Copenhagen, 2200 Copenhagen, Denmark; jrekling@sund.ku.dk; 6Cell Therapy R&D, Novo Nordisk, 2760 Måløv, Denmark; carlosq825@gmail.com (C.D.Q.); sqyf@novonordisk.com (S.E.D.F.); charlottevbertelsen@gmail.com (C.V.B.); carlos.villaescusa@gmail.com (J.C.V.); 7Target Discovery Platforms, Novo Nordisk, Seattle, WA 98109, USA; aomc@novonordisk.com (A.J.M.); acoa@novonordisk.com (A.C.); 8GOLD In Vivo Pharmacology DK I, Novo Nordisk, 2760 Måløv, Denmark; mikg@novonordisk.com (M.K.G.); jlnu@novonordisk.com (L.M.J.)

**Keywords:** neuropeptide FF receptor 2, neuropeptide FF, screening platform, human-arcuate-like neurons, appetite control, obesity

## Abstract

Restoring the control of food intake is the key to obesity management and prevention. The arcuate nucleus (ARC) of the hypothalamus is extensively being studied as a potential anti-obesity target. Animal studies showed that neuropeptide FF (NPFF) reduces food intake by its action in neuropeptide Y (NPY) neurons of the hypothalamic ARC, but the detailed mode of action observed in human neurons is missing, due to the lack of a human-neuron-based model for pharmacology testing. Here, we validated and utilized a human-neural-stem-cell-based (hNSC) model of ARC to test the effects of NPFF on cellular pathways and neuronal activity. We found that in the human neurons, decreased cAMP levels by NPFF resulted in a reduced rate of cytoplasmic calcium oscillations, indicating an inhibition of ARC NPY neurons. This suggests the therapeutic potential of NPFFR2 in obesity. In addition, we demonstrate the use of human-stem-cell-derived neurons in pharmacological applications and the potential of this model to address functional aspects of human hypothalamic neurons.

## 1. Introduction

Food intake regulation is a major target for anti-obesity treatment strategies. The anorexigenic properties of neuropeptide FF (NPFF) in rodents and chicks have been long known [1,2,3,4]. It has also been recently discovered that NPFF receptor 2 (NPFFR2) is also a crucial regulator of diet-induced adaptative thermogenesis in mice [5,6].

NPFF, also known as morphine-modulating neuropeptide, is highly expressed in the appetite-regulating centers of the brain, such as the hypothalamus and pons medulla, as well as in the pituitary and spinal cord [7,8,9]. NPFF primarily acts on two Gi/Go-associated G-protein-coupled receptors: NPFFR1 (GPR147) and NPFFR2 (GPR74). However, NPFF has a higher affinity and potency for NPFFR2 [10,11,12]. While NPFFR1 is ubiquitously expressed in the CNS and has been associated with regulation of stress response and reproduction [12,13,14,15,16], NPFFR2 is expressed in distinct brain regions and neurons, including the neuropeptide Y (NPY) neurons of the arcuate nucleus (ARC) of the hypothalamus, positioning NPFFR2+ NPY neurons to play a key role in food intake, appetite modulation and overall, in metabolism [6,16]. In addition, NPFFR2 is expressed in the paraventricular nucleus of the hypothalamus (PVN) and at low levels in the superficial layers of the spinal cord and thalamic nuclei [11,16,17,18]. Indeed, hypothalamic NPY neurons play a pivotal role in food intake and interact with a multitude of other neurons [19]. For instance, NPY neurons from the ARC project to corticotropin-releasing hormone (CRH) neurons of the PVN, and release of NPY inhibits CRH neuron activity through Gi-coupled (inhibitory) receptor Y1 [19,20,21]. Thus, applying a Gi agonist such as NPFF to inhibit ARC NPY neurons would dis-inhibit PVN CRH neurons [19,20,21]. Accordingly, the NPFFR2-mediated effect of NPFF on food intake and energy balance is believed to be due to increased neuronal activity in the PVN, modulating signaling along the hypothalamo–pituitary–adrenal (HPA) axis and the autonomic nervous system [6,16,22,23]. As shown in in vivo mouse studies, an intracerebroventricular (i.c.v.) injection of NPFFR2 agonist dNPA in mice leads to the activation of neurons in the PVN and a subsequent increase in corticosterone levels in a CRH-dependent manner [22]. In support of this mechanism, as predicted in animal studies, activation of NPFFR2 has been shown to inhibit cAMP levels in ARC neurons in mouse studies. However, no studies so far demonstrate how this inhibition of cAMP changes affects the activity of the NPFFR2+ neuronal population in the ARC and whether this effect can be observed in human neurons. 

Our understanding of food intake control vis-à-vis the HPA axis in weight gain in humans is still limited. Viable primary human hypothalamic neurons for biological studies are not readily available, and existing studies on the action of NPFFR2 in a human in vitro model have been performed in a transfected tumor cell line [24]. Therefore, a more physiologically relevant, human-based model of appetite-controlling neurons is needed. Additionally, such a model would allow researchers to perform studies on signaling mechanisms of NPFF to gain new insights into the nature of anorexigenic effects of NPFFR2 and to evaluate the therapeutic potential of NPFFR2 as an anti-obesity target. Fortunately, recent advances in the stem cell field have allowed us to develop a sophisticated in vitro model based on human embryonic (hESC) or induced-pluripotent-stem-cell (iPSC)-derived hypothalamic neurons [25,26,27,28,29,30]. While these neurons do not fully resemble the neuronal population of the ARC, this cell-based platform can be validated to address specific research questions that can be applied for human-cell-based assays in therapeutic target discovery, toxicology and drug development [31,32,33], as well as in vitro disease modeling using patient-iPSC-derived neurons [34].

Here, we used human hypothalamic ARC-like neurons (hALN) derived from hESCs to study intracellular effects of NPFF and its control of functional activity of the neurons based on their calcium responses. Our study shows that NPFF stimulation abolishes forskolin-induced cAMP increases and reduces calcium activity in human neurons, confirming same mode of action for NPFF in human neurons as described for the rodent model [35] and nonhuman in vitro models [11,17,24].

## 2. Results

### 2.1. Characterization of NPFFR2 Neuron Population

The presence of *NPFFR2* expression in the human ARC has previously been reported [36,37], but we aimed to characterize the neuronal population expressing *NPFFR2* in the ARC in order to identify specific markers for these neurons. Because the effects of NPFFR2 activation were reported in mice, and because single-cell (SC) gene expression data from human hypothalamic neurons suitable for reanalysis were not available from published sources, we first performed reanalysis of previously published SC gene expression data from mouse hypothalamic neurons (see Section 4.1). The expression pattern of NPFF receptors was analyzed in four studies [38,39,40] covering different subsets of murine hypothalamic nuclei (Figure 1A). In the reanalysis, we found that *Npffr2+* cells were most abundant in the datasets that contained the highest percentage of ARC neurons [38,41] (Appendix A), thereby confirming observations previously published by Zhang et al. [6]. Analysis of neuronal subclusters in the study by Campbell et al. [38], which contained data exclusively from the ARC, showed that *Npffr2+* neurons are predominantly *Npy*+− (64.5%), *Agrp*+− (59.9%), and glutamate-decarboxylase-1 and 2 (*Gad1* and *Gad2*)-expressing GABAergic neurons (63.8% and 64.5%, respectively). A smaller subset of *Npffr2+* ARC neurons were *Slc17a6*-expressing glutamatergic or dopaminergic (*Ddc+*) neurons (21.7% and 26.3%, respectively) (Appendix A). In addition, some *Npffr2+* neurons also expressed low levels of proopiomelanocortin (*Pomc*) and somatostatin (*Sst*) (40.1% and 28.3%, respectively) (Appendix A). A small percentage of *Npffr2+* neurons expressed *Npffr1* and nociceptin (*Pnoc*) (16.4% and 15.8%, respectively). *Npffr1* was expressed in a larger number of neurons, but these two receptors were generally not colocalized (Appendix A) [38,39,40,41], and the *Npffr1* expression level in *Npffr2+* neurons was low (0.13 ± 0.02 SEM; Figure 1B). Gene expression analysis confirmed that *Npffr2+* neurons co-express *Npy* and *Agrp* in relatively high abundance (3.02 ± 0.21 and 2.31 ± 0.17; Figure 1B), and GABAergic, glutaminergic and dopaminergic genes to a lower degree, as shown in Figure 1B and Appendix A. Other enriched genes in *Npffr2+* neurons are presented in Figure 1C and Appendix A. Thus, the majority of *Npffr2*-expressing neurons in mouse ARC were *Npy/Agrp+* neurons. 

We next performed an in situ hybridization labeling for *Mm-Npffr2* in mouse hypothalamus and confirmed that abundant expression of *Mm*-*Npffr2* was detected in the ARC (Figure 1D), and the proximity of this signal to the third ventricle (3V) suggests that these cells were likely *Npy+/Agrp+* neurons. To test the co-expression of *Hs-NPFFR2* with *Hs-NPY*, *Hs-AGRP*, *Hs-VGAT* and *Hs-POMC* in human hypothalamic neurons, we performed double in situ hybridization labeling with these markers in human postmortem brain tissue (Figure 1E). The expression of *NPFFR2* in the human ARC (ventral portion) was less abundant than in the mouse arcuate nucleus, but we detected expression of this receptor in *NPY-*, *VGAT-* and *AGRP*-positive neurons but not in *POMC*-positive neurons (Figure 1E), which was consistent with the *NPFFR2-*expressing population described in the mouse ARC based on our SC gene expression reanalysis.

### 2.2. Human-Stem-Cell-Derived ARC Neurons Resemble NPFFR2-Postive Neuron Population in the Hypothalamus

To study the effect of NPFF signaling on the neuronal activity of ARC, we developed an in vitro model of human hypothalamic ARC neurons. This model was based on hESC-derived neural stem cells (further referred to as hNSC, patent no. WO2021004864A1). In brief, hypothalamic differentiation was first induced by a combination of morphogens, and the differentiating hypothalamic neurons (hDHN) were matured for 20 days, resulting in a population of neurons resembling ARC. The hESC-derived neural stem cells’ differentiation and maturation into human hypothalamic ARC-like neurons (further referred to as hALN) was based on earlier published protocols [25,26,27,34] with some modifications (see Section 4.2 and Appendix A). To test for successful induction of hypothalamic differentiation in our experiments and to characterize the hALN, we compared gene expression of specific hypothalamic markers at the three stages of the hALN generation: nondifferentiated hNSC, differentiated hDHN and matured hALN.

hNSCs are characterized by the expression of transcription factors SOX1, PAX6 and POU5F1 [25]. In alignment with these previous data, our hNSCs also exhibited robust expression of these genes (Appendix A). Hypothalamic differentiation was confirmed by an increase in the expression of transcription factors NKX2.1 (Figure 2A) necessary for driving the differentiation of NSC into ARC hypothalamic neurons, in particular, for NPY/AgRP neurons [25,41,42,43,44,45]. In contrast to previously published data [25,43,44], we did not detect an increased expression of RAX in hDHNs (Figure 2A and Appendix A), which could be due to the fact that RAX is transiently expressed during hypothalamic development. However, the late progenitor marker of ARC neurons, DBX1, was upregulated in hDHNs compared to hNSCs (Figure 2A and Appendix A). Among other hypothalamic markers, OTP and SIM1, driving the differentiation of NSC into the paraventricular nucleus of the hypothalamus [44], were strongly upregulated in hDHNs compared to hNSCs and increased further during the maturation into hALNs (Figure 2A). We also tested the nonhypothalamic markers FOXG1 and EMX1, driving telencephalon development, and EN2, a marker for the midbrain and hindbrain. We found that they were very low in abundance in hALNs compared to hNSCs or not detected at all (Figure 2A and Appendix A).

Next, we tested neuropeptide expression in the hNSC cultures as one of our main criteria for functional maturity in these stem-cell-derived neurons. Expression of *NPY*, GABAergic markers (GAD1 and GAD2), *SST*, dopaminergic neuron marker tyrosine hydroxylase (*TH*) and glutaminergic marker (*SLC17a7)* was increased in hALNs compared to immature hNSCs and hDHNs (Figure 2B). *CARTPT* mRNA level was detected in hDHNs and hALNs, but not hNSCs (Appendix A). *POMC* expression levels were low in all three groups (Figure 2B and Appendix A). More importantly, the expression of *NPFFR2* increased in hALNs (Figure 2B).

We next performed immunostainings for neuropeptides and other neuronal markers in hALNs. All 20-day matured hALNs were positive for neuronal markers of dendrites (MAP2), synaptic vesicles (synaptophysin, SYP) and the neuronal cytoplasmic marker HuC (Figure 2C). Ninety-four percent of hALNs were immunopositive for NPY (94.83% ±1.01 SEM; Figure 2C), 71.21% (±3.15) of all cells were also positive for GABA (marked by Vgat expression), and 50% (±4.26) neurons showed several AGRP-immunoreactive puncta per cell (Figure 2C). NPFFR2 was present in the majority of hALNs (Figure 2C). POMC neurons were only detected in cultures matured for longer than 28 days, and their number was about 0.1% (±0.003) (Figure 2C). We also detected tyrosine-hydroxylase (TH)-positive cells (about 10%), demonstrating that some neurons differentiated into dopaminergic neurons, and also single CRH-positive cells comprising less than 0.1% of the population (Figure 2C). 

A cell population consisting primarily of one cell type (in this case, NPY neurons in the matured hALN population) is likely to display a more homogenous response to drugs and therefore has an advantage over heterogeneous populations for pharmacology assays based on readings from the whole well. Considering this, we performed cAMP response analyses on 20-day-matured hALNs, when the majority of cells expressed NPY, GABA and NPFFR2 (Figure 2C).

### 2.3. Human-Stem-Cell-Derived ARC Neurons Display Action Potentials and Calcium Oscillations

To test for electrophysiological maturity in hALNs, we used whole-cell patch recordings to record action potentials between 21 and 25 days of maturation (Figure 3A,B). Neurons had a large input resistance of 3.5 ± 1.9 GOhm and responded to short depolarizing pulses with brief trains of action potentials. Longer pulses resulted in spike inactivation, but the neurons could be driven to repetitive spiking with short (10 ms, 200 ms interval) repetitive trains.

Although electrophysiological tests are the gold standard in studies on neuronal activity, they cannot be applied in a screening format, which makes evaluation of drugs based on changes in membrane potential, i.e., neuronal firing, difficult. Neurotransmitter release is governed by membrane changes driving an increase in cytoplasmic and synaptic calcium concentration [46,47], and therefore intracellular calcium measurements can be used as an indirect assessment of neurotransmitter/neuropeptide release. Fluorescent calcium dye allows detection of neuronal activity as a high throughput method with sufficient temporal resolution to determine the effects of different compounds. Therefore, we used live-cell calcium imaging as a readout for neuronal activity in our hALNs. 

We first determined the effective dose of NPFF required to activate NPFFR2 in hALNs based on a cAMP assay in the presence of 300 nM forskolin. Activation of NPFFR2 inhibits adenylate-cyclase in cells [35,45], and in our experiments NPFF strongly inhibited a forskolin-induced cAMP increase in hALNs, with a half-maximal effective concentration (IC50) of 0.1 ± 0.01 nM (Figure 3C). For comparison, the hNSCs showed no response to NPFF (Figure 3C).

To test the effect of NPFFR2 on the cytoplasmic calcium dynamics, we measured intracellular calcium concentrations using hALNs loaded with Fluo-8. Typically, neurons display calcium oscillations with synchronized patterns, as cells become mature and capable of forming electrical connections with one another [26,48]. We first recorded basal spontaneous activity in 21–30-day old hALNs and observed synchronized calcium oscillations in 30–70% of cells (Figure 3D–F). Stimulation with 50 mM KCl induced a cytoplasmic calcium transient increase (Appendix A).

We next recorded calcium responses to 10 nM NPFF to maximally activate NPFFR2 [35,45] in hALNs. NPFF reduced the frequency of oscillations (Figure 3D) and reduced calcium levels (in nonoscillating cells) in 78 ± 8% (Figure 3E, F, data from five experiments with 13–22 cells per experiment). In nonoscillating cells, NPFF decreased Fluo-8 intensity by 31 ± 5% (*p* < 0.001 compared to control). In 8 ± 3% cells, we detected no change, and the remaining 13 ± 5% cells increased their oscillation rate (n = 12). As cytoplasmic calcium changes control neurotransmitter release, the decreased rate of calcium oscillations in the majority of hALNs implies inhibition of NPY neuron signaling by NPFF [46,47]. In addition, we showed that hALNs are a sensitive platform for receptor–ligand interaction, as shown with NPFF and NPFFR2, and for mode of action testing.

## 3. Discussion

The modulation of brain appetite centers is a primary mechanism targeted by modern obesity therapies and in development of next-generation anti-obesity drugs. Here, we show that NPFF reduces cAMP levels and decreases oscillatory activity in human ARC NPY neurons, which mediate the orexigenic effect in human hypothalamus. In addition, we characterized in vitro maturation of hNSC into ARC-like predominantly gabaergic NPY neurons, establishing the relevance of this platform for NPFF testing, and demonstrated their physiological activity with both electrophysiology and calcium imaging.

As it has been previously shown that *Npffr2* is predominantly expressed in GABAergic *Npy/Agrp* neurons in the ARC of the hypothalamus in mice, here we started with the reanalysis of published data to characterize this specific neuronal population with regard to other markers. Thus, the reanalysis showed that a significant part of *Npffr2+* neurons also expressed *Pomc* and *Pnoc*, although at very low levels, which has not been reported previously. These neurons are believed to be a distinct population from Npy/Agrp neurons. Unfortunately, the number of identified *Npffr2-*expressing neurons in the mouse ARC [38] was too low to allow for reliable analysis of different subgroups based on their neuromediator expression. A low number of detected *Npffr2+* neurons may be explained by low mRNA cycling of GPCRs in general, compared, for example, to neuropeptide expression [49]. For comparison, ghrelin receptor *Ghsr*, which is thought to be expressed in the majority of *Npy/Agrp* neurons, was detected in only 18% of cells in the study by Campbell et al. [38]. On the other hand, the in situ hybridization labeling showed an abundance of *Npffr2+* neurons in the mouse ARC. The presence of *NPFFR2* was also detected in the human ARC, in neurons expressing *GABA*, *NPY* and *AGRP*, but not in neurons expressing *POMC*, as shown by the in situ hybridization labeling. This suggests a plausible role for NPFFR2 in food intake control and analogous NPFFR2-induced cellular pathways in this population of human ARC neurons, which confirms the relevance of testing NPFFR2 signaling in a human-based model of ARC.

To directly probe the function of NPFFR2 in human hypothalamic neurons, we employed an hESC-derived model of human hypothalamic neurons. While human-stem-cell-derived ARC-like neurons have been previously described [25,26,27,34], the use of these cells in pharmacology research has been limited as the characterization of receptor expression in this platform and its similarity to native human ARC neurons has not been established. hALNs do not mimic primary human neurons in every aspect as metabolic conditions and gradients of differentiation cues during the embryonic development cannot be reproduced in vitro. However, based on the expression of main early and late hypothalamic development markers in hDHN, our protocol efficiently induced hypothalamic differentiation in human neuronal stem cells, and further maturation of these cells increased expression of main ARC neuropeptides *NPY*, *AGRP* and *POMC* at later stages to detectable levels for immunostainings, as described in the literature [27,34,44]. Moreover, hALNs displayed action potentials and repetitive train behavior, demonstrating the ability of burst firing, which is a well-described electrophysiological characteristic for arcuate NPY neurons [46,47]. hALNs also exhibited spontaneous synchronized intracellular calcium oscillations characteristic of primary neurons [50]. Therefore, hALNs is an appropriate platform with which to study physiological effects of relevant stimuli in vitro, which may inform the in vivo effects of this biology in the human hypothalamus. Based on earlier studies, the degree of heterogeneity normally observed in ARC, such as the presence of POMC and other neuronal populations, can be increased by, for example, addition of astrocyte-conditioned medium during the maturation phase [51] or incubation with leptin [26]. In this study, we did not apply these conditions as we viewed the relative homogeneity of the population (>90% NPY neurons) as an advantage for pharmacological assays for studies on the effect of NPFFR2 activation.

Our study is the first to demonstrate the expression of NPFFR2 by immunostaining in the majority of NPY-positive hALNs and increased expression of *NPFFR2* as cells matured. Critical to the utilization of our model for assessing in vitro pharmacology, we found that NPFF activated NPFFR2 in hALNs, resulting in an inhibition of forskolin-induced cAMP levels. This action in human neurons was consistent with the mode of NPFFR2 signaling (Gi-coupled receptor) and with NPFFR2 action in mouse neurons reported previously [24]. Unlike routine in vitro assessments that rely on induced receptor overexpression, the detected cAMP response in our hALN experiments relied solely on the endogenous NPFFR2. Nevertheless, cAMP responses were detected in our model even at lower NPFF concentrations than those reported in transfected cells [24]. Additionally, calcium measurements demonstrated that NPFFR2 reduced neuronal activity in hALNs, as shown by the reduced rate of calcium spikes. This showed a high sensitivity in our platform for interrogating NPFF and NPFFR2 biology and illustrated the use of the model for screening compounds that may have direct human relevance.

While our data show that NPFFR2 can be targeted pharmacologically in a human-based system, the use of NPFFR2 agonists for obesity treatments may have a number of limitations. It is difficult to predict whether the following signaling cascade along the HPA axis would result in the reduction in food intake in human, as described in mice. Increased CRH signaling in the chronic setting in humans is often associated with dysregulated signaling along the HPA and increased cortisol levels, promoting food intake [52,53]. On the other hand, cortisol is also described as a catabolic hormone in the acute setting and, in concert with the release of catecholamines, can induce anorexigenic responses and mobilization of energy stores [54]. In obesity, sympathetic tone is increased due to hyperleptinaemia, and activation of the HPA axis may further elevate it, resulting in increased blood pressure and tachycardia [55]. In the development of agonists or antagonists to NPFFR2, the balance between the acute and chronic stress responses and the tissues engaged must be considered. Thus, the hALN cellular assay could be used as a tool for initial evaluation of optimal and selective NPFFR2 compounds prior to further testing in vivo.

Collectively, our study reveals the effect of NPFF on ARC NPY neuron activity in a human-based system, indicating the potential of targeting NPFFR2 for anti-obesity therapies. We also conclude that hALN is a highly sensitive platform for studies on ligand–receptor interactions and neuronal activity assessments, which enabled us to detect the NPFF-induced signaling pathway from NPFFR2 receptor activation to inhibition of NPY neuron activity. This demonstrates that hALNs can be used to address specific research questions based on the presence of receptors and neuromediators, which will likely help us to predict more precisely the mode of action and efficacy of developing drugs in the human hypothalamus. Importantly, establishing differentiation protocols to mimic various hypothalamic areas and characterization of cells will accelerate the development of drugs targeting the brain and help us to further understand the signaling mechanisms in the human hypothalamus.

## 4. Materials and Methods

### 4.1. Reanalysis of Single-Cell RNA Sequencing Data from Published Sources

Single-cell RNA-seq gene expression matrices were downloaded from the Gene Expression Omnibus datasets GSE93374 [38], GSE125065 [40] and GSE74672 [39] and from Mendeley Data dataset [41]. Each dataset was processed separately using the SCTransform-based workflow from the Seurat R package [56]. For the Campbell et al. data, neurons from all feeding conditions were used and identified via the author-provided metadata table. For the Mickelsen et al. data [40], both samples were used, and neurons were identified based on the markers Slc17a6, Slc32a1 and Snap25. For the Romanov et al. data [39], neurons were identified based on the author-provided metadata table column “level1 class”. Finally, for the Kim et al. data [41], neurons from all behavioral conditions were used and identified as in in the Mickelsen et al. data [40]. A neuron was found to express a gene if it had >0 counts of the gene. To adjust for the number of genes expressed per cell, which depends on the quality of the dataset, we calculated an “adjustment factor” (Appendix A). First, the average number of genes expressed per cell (n/cell) was calculated for each dataset. The Campbell adjustment factor was set to 1, as it had the fewest genes per cell. The factor of the other datasets was calculated by dividing n/cell from Campbell with n/cell from each of the other datasets. For the differential expression analysis between Npffr2+ and Npffr2- neurons in the Campbell data, the Wilcoxon rank-sum test (Seurat R package) was used with min.pct = 0.1, min.pct.diff = 0.1, log fold change threshold > 0.25 and adjusted *p* value < 0.05.

### 4.2. Generation of Human-Embryonic-Stem-Cell-Derived Hypothalamic Neurons

Human embryonic stem cells (hESCs) were differentiated into hNSCs following the patent application WO2021004864A1. These cells were *Nestin*-*, Pax6*-*, SOX2*- and *OTX2*-positive and formed neural rosettes, which is a common signature of neural stem cells. The directed hypothalamic-like differentiation was performed by activating SHH signaling for ventralization and inhibiting telencephalic development by Notch inhibition using a 7-day differentiation protocol (Appendix A).

hNSC were detached using TrypLE™ (#12604013—Gibco™, Bleiswijk, The Netherland) and after collection, they were treated with Defined Trypsin Inhibitor (#R007100—Gibco™, Bleiswijk, The Netherland) and centrifuged at 300× *g*. Cells were then resuspended in basic medium containing DMEM/F12 with Glutamax (#10565018, Gibco™, Bleiswijk, The Netherland), 20 U/mL Penicillin-streptomycin (Gibco™, Bleiswijk, The Netherland), 1% CTS™ N2 (#A1370701, Gibco™, Bleiswijk, The Netherland) and B27 (XenoFree without vitamin A #A3353501, Gibco™, Bleiswijk, The Netherland) supplemented with 10 µM ROCK inhibitor (Y-27632 dihydrochloride, #1254, Tocris, Bristol, UK) and subsequently centrifuged. Then, cells were resuspended in basic medium supplemented with 500 ng/mL recombinant human Sonic Hedgehog (SHH—#1845-SH-025/CF, R&D systems, Abingdon, UK), 400 nM InSolution Smoothened Agonist (SAG—#566661-500UG, Sigma-Aldrich, Søborg, Denmark) and 100 ng/mL Human FGF8b (#130-095-738, Miltenyi Biotec, Bergisch Gladbach, Germany) and seeded at 1.5 × 10^5^ cells/well in a 48-well plate precoated with mouse Laminin (1.5 µg/cm^2^—#114956-81-9, Merck, Mannheim, Germany) and maintained for 3 days. On day 4, medium was replaced with basic medium supplemented with 10 µM DAPT (#2634, Tocris, Bristol, UK)) for 4 days. After day 7, the hDHN were cryopreserved and stored at −150 °C. Upon thawing, the cells were plated into 96-well plates (Perkin Elmer, Germany) in basic medium supplemented with 20 ng/mL BDNF (#SRP3014-10UG, Sigma-Aldrich, Søborg, Denmark), 20ng/mL GDNF (#450-10, PeproTech, Cranbury, UK) and 10 µM DAPT (#2634, Bio-Techne, Bristol, UK) for 3 days, after which DAPT was withdrawn from the medium. From day 10 onwards, the cells were further matured for about 20 days with replacement of 50% of medium with fresh medium every 2 days. 

### 4.3. Immunostainings

Cells were fixed in the wells with 4% PFA for 30 min at room temperature and washed 3 times with PBS. The cells were permeabilized with 0,15% Triton X with 2% bovine serum albumin and primary antibodies to NPY (#ab30914, Abcam, Cambridge, UK), Vgat (MA5-24643, Invitrogen, Bleiswijk, The Netherland), synaptophysin (#A6442, Invitrogen, Bleiswijk, The Netherland), NPFFR2 (#NB300-169, Novus Biologicals), POMC (#ab73092, Abcam, Cambridge, UK), AgRP (#A1059-62U, Nordic Bio Site), TH (#ab137869, Abcam, Cambridge, UK), CRH (#bs-0382R Bioss), MAP2 (#M4403, Sigma-Aldrich, Søborg, Denmark) and HuC (#A21271, Invitrogen, Bleiswijk, The Netherland) applied overnight. After a 3-time wash with PBS, the secondary antibodies (Alexa fluor 488 and 568, 1:400; Invitrogen, Bleiswijk, The Netherland) and DAPI (Hoechst 33342 Solution, #62249, Thermo Fisher Scientific, Bleiswijk, The Netherland) were added for 1 h at room temperature. Images of immunostainings were produced using fluorescent microscope IX81 Olympus (Tokyo, Japan) with Olympus CellSens Dimension 2.3 software (Olympus Corporation, Tokyo, Japan) and analyzed with ImageJ software (NIH, Bethesda, MD, USA). For quantification of NPY, AGRP, GABA and POMC cell numbers, costaining with synaptophysin was used to determine the total number of neurons.

### 4.4. In Situ Hybridization

ISH on human and mouse brain sections was performed as described previously [57], with modifications. Formalin-fixed, paraffin-embedded (FFPE) blocks containing either mouse brains or human hypothalamus (procured from the Edinburgh Brain Bank, SD010/17, BBN001.29880) were sectioned at 5 μm onto Fisher SuperFrost Plus glass (Fisher Scientific, Roskilde, Denmark). Mouse sections were hybridized with a rodent-specific probe to detect mouse *Npffr2* mRNA (#410178-C4, Advanced Cell Diagnostics, Bio-Techne, Bristol, UK). Human hypothalamus sections containing arcuate nucleus were hybridized with a human-specific probe to detect mRNA transcripts for *NPFFR2* (#834298 and #834298-C2) and multiplexed with human specific probes for either *AGRP, vGAT* (SLC32A1), *NPY* or *POMC* (#557458, #415688, #416678, #429908, Advanced Cell Diagnostics, Bio-Techne, Newark, CA, USA). Multiplex FISH was performed using the Leica RX Fully Automated Research Stainer (Leica, Ballerup, Denmark) and amplified/stained using the RNAscope LS multiplex fluorescent assay kit (Advanced Cell Diagnostics, Bio-Techne, Newark, CA, USA) and Opal fluorophore reagent packs (Akoya BioSciences, Inc., Marlborough, MA, USA). All slides were counterstained with DAPI and were coverslipped with either EcoMount (BioCare, Pacheco, CA, USA) or ProLong Diamond antifade mountant (ThermoFisher Scientific, Bleiswijk, The Netherland). Mouse brain sections were imaged at 20× on a TCS SP8X confocal microscope (Leica, Ballerup, Denmark). Post hoc processing matched brightness/contrast across all slides, and images were compiled in Adobe Illustrator (Adobe Inc., San Jose, CA, USA) for presentation. Whole human hypothalamus sections were scanned with the VS200 slide scanner using (Olympus, Tokyo, Japan) a 40× air objective (0.95 NA) and a DAPI/CY3/CY5 filter set. Images were prepared with the Olympus ASW software, and signal intensity levels were adjusted to match across staining/slides.

### 4.5. Gene Expression

RNA was purified using RNeasy Micro Kit (Qiagen, Austin, TX, USA) according to manufacturer’s protocol. cDNA was synthesized using iScript™ Reverse Transcription Supermix (Bio-rad, Herlev, Denmark). Quantitative real-time PCR was carried out on ViiA-7 using 1 ng of cDNA mixed with TaqMan™ OpenArray™ Real-Time PCR Master Mix (Applied Biosystems™, Thermo Fisher Scientific, Bleiswijk, The Netherland) and TaqMan primers (Appendix A). Gene expression was analyzed by the 2 ΔCT method with beta actin (ACTB) as endogenous control.

### 4.6. cAMP Assay

For cAMP assays, hALNs were grown in 96-well plates for 21–25 days as described above and incubated with 300 nM forskolin in culture medium for 30 min, and then NPFF (#3137, Tocris, Bristol, UK)) at various concentrations was added to respective wells for additional 30 min. After that, the cells were lysed with CisBio lysis buffer, and cAMP detection was carried out with CisBio reagents (cAMP Gi kit, # 62AM9PEC, CisBio, Perkin Elmer, Germany) according to manufacturer’s instructions. A small number of lysed cells was reserved for ATP measurements (ATPlite, # 6016736, CisBio, Perkin Elmer, Germany) to normalize the cAMP readings to number of cells in wells. The plate was read on a Mithras reader, and GraphPad Prism was used to perform the calculations and generate the graphs.

### 4.7. Electrophysiological Recording

Electrophysiology: Neurons plated on glass coverslips were placed in a recording chamber superfused with an artificial cerebrospinal fluid (aCSF) solution containing (in mM): 138 NaCl, 4.5 KCl, 1.2 NaH_2_PO_4_, 25 NaHCO_3_, 1.0 D-Glucose, 1.2 MgCl_2_, 2.6 CaCl_2_ and 10 HEPES (all from Sigma-Aldrich, Søborg, Denmark, RRID:SCR_008988), with a pH of 7.4, equilibrated by bubbling with 95% O_2_/5% CO_2_. Glass micropipettes were pulled from filamented capillary glass (O.D. 1.5 mm, I.D. 0.86 mm, Harvard Apparatus, Holliston, MA, USA) using a PUL-100 micropipette puller (World Precision Instruments, Hessen, Germany, RRID:SCR_008593) with a tip resistance of 4–6 MΩ. Patch pipettes were filled with a solution containing (in mM): 130 HCH_3_SO_3_, 130 KOH, 10 HEPES, 0.4 NaGTP, 4 Na_2_ATP, 5 Na_2_-phosphocreatine and 4 MgCl_2_ (all from Sigma-Aldrich, Søborg, Denmark, RRID:SCR_008988). Patch pipettes were visually guided to target neurons under visual control using MPC-200 micromanipulator system (Sutter Instruments, Novato, CA, USA) on a fixed-stage upright microscope (modified Olympus BX51, Olympus Corporation, Tokyo, Japan) under 40× magnification (NA = 0.8, WD = 3.3 mm). Somatic whole-cell patch-clamp recordings were performed in current clamp using an AxoClamp 2B amplifier (Molecular Devices, Sunnyvale, CA, USA), and the data were digitally acquired at a sampling rate of 5 kHz, with a low-pass filter of 2 kHz. Trains of stimulation pulses (10 ms duration) were given at current levels just high enough to elicit a single AP per pulse.

### 4.8. Cytoplasmic Calcium Measurements

Cytosolic (Ca2+)i was measured using Ca^2+^ indicator Fluo-8. Cells were loaded with 1 µM Fluo-8 (Calcium Flux Assay Kit, #ab112129, Abcam, Cambridge, UK) for 30 min according to manufacturer’s protocol. Recording of spontaneous calcium activity was performed prior to NPFF challenge (time 0). Baseline signal was measured at 488 nm excitation and 510 nm emission using inCell analyzer 2000 for 5 min. After that, 10 nM NPFF (#3137, Tocris, Bristol, UK)) was added to the wells following another 5 min of Fluo-8 signal recording from the same field. As control, same amount of medium without NPFF was added in similar recordings. No significant changes in Fluo-8 signal intensity pattern were observed in control wells compared to the baseline. After the background subtraction, calcium responses were processed and analyzed using ImageJ. Mean intensity over time was measured in ROIs including cytoplasmic soma area of individual cells and their traceable extensions. The decrease in nonoscillating cells was calculated as the lowest value below the baseline derived from averaging the Fluo-8 intensity during the last minute before the NPFF challenge. Data are presented as fluorescence intensity in arbitrary units.

### 4.9. Quantification and Statistical Analysis

Statistical analyses were performed using Student’s two tailed unpaired t-test for calcium oscillation analyses, n represents number of analyzed cells, and values are given as SEM. Statistical analyses were performed using mixed-effects model (REML) analysis and post hoc Tukey’s multiple comparison test analysis for comparison of gene expression. n represents cells lysed at a defined time point. Statistical analyses were performed using GraphPad Prism 9.0.1.

## Figures and Tables

**Figure 1 ijms-23-03260-f001:**
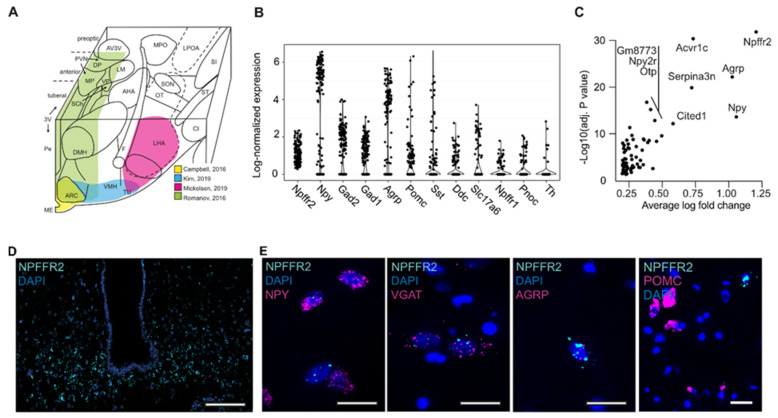
Characterization of *NPFFR2-*positive neuronal population in mouse and human hypothalamus (**A**) 3D schematic representation of the hypothalamus with the single-cell RNAseq-analyzed regions highlighted. Rostral-to-caudal view of major hypothalamic nuclei in the right hemisphere of rat hypothalamus. Abbreviations: AHA, anterior hypothalamic area; ARC, arcuate nucleus; AV3V, anteroventral area of third ventricle; CI, capsula interna; DP, dorsal parvocellular subnucleus of paraventricular nucleus (PVN); DMN, dorsomedial nucleus; F, fornix; LHA, lateral hypothalamic area; LM, lateral magnocellular subnucleus of paraventricular nucleus; LPOA, lateral preoptic area; ME, median eminence; MP, medial parvocellular PVN; MPO, medial preoptic area; OT, optic tract; SCh, suprachiasmatic nucleus; SON, supraoptic nucleus; SI, substantia inomminata; ST, subthalamic nucleus; VMN, ventromedial nucleus; VP, ventral parvocellular subnucleus of paraventricular nucleus. This schematic diagram is based on a previously published article from Berthoud et al. [4]. (**B**) Expression level of specific neuronal markers in *NPFFR2+* neurons (Data are shown as means ± SEM), data are from Campbell et al., 2016 [38]. Due to low number of *NPFFR2+* neurons in mouse ARC, all groups were pooled for this analysis. (**C**) Significantly enriched genes in *Npffr2+* neurons (Data are shown as means ± SEM). (**D**) In situ hybridization staining for *NPFFR2* in adult mouse hypothalamus. *NPFFR2* mRNA (cyan) was detected in the ARC, primarily in cells with proximity to the ventral portion of the 3rd ventricle. Panel inset demonstrates the detail of *Npffr2* expression in *Npffr2+* cells in the ARC. Scale bars = 200 µm; inset = 50 µm. (**E**) In situ hybridization staining for *NPFFR2* and *NPY*, *GABA*, *AGRP*, or *POMC* in human arcuate nucleus (ventral portion) shows co-expression of *NPFFR2* with NPY, GABA and AGRP but not POMC. Scale bars = 10 µm.

**Figure 2 ijms-23-03260-f002:**
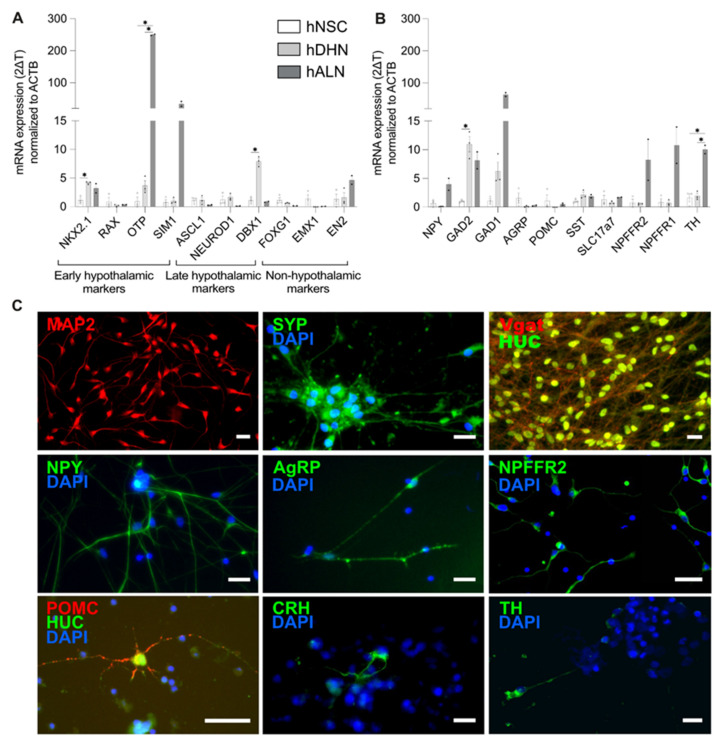
Differentiated and matured hALNs show markers of human ARC neurons. (**A**) Gene expression of early and late hypothalamic progenitor markers, and nonhypothalamic markers (n = 2–3, data are shown as means ± SEM; * *p* < 0.05), and (**B**) gene expression of neuropeptide markers and in hNSC, hDHN, hMHALN and hALN (n = 2–3, data are shown as means ± SEM; * *p* < 0.05). (**C**) Immunostainings of hALNs for (from left to right and top to bottom) MAP2, synapsin (SYN), Vgat and HUC, NPY, AgRP, NPFFR2, POMC and HUC, CRH and TH (scale bar 20 µm).

**Figure 3 ijms-23-03260-f003:**
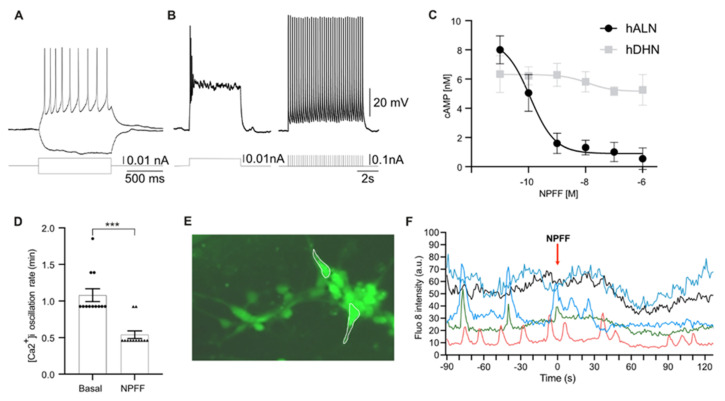
Characterization of the hALN. (**A**) 1 s-long depolarizing, and hyperpolarizing pulses injected into a hALN neuron. (**B**) Left panel: 5 s-long depolarizing pulse leading to spike inactivation; right panel: train of 10 ms pulses (200 ms interval) inducing repetitive action potentials (n = 8); (**C**) Effect of NPFF on forskolin-induced increases in cAMP in hNSC (gray) and hALN (black) (n = 3, data are shown as means ± SEM). (**D**) (Ca^2+^)_i_ oscillation rate (min) in basal condition and after 10 nM NPFF stimulation in hALN (n = 12; *t*-test *** *p* < 0.000, data are shown as means ± SEM). (**E**) Representative image of Fluo-8-loaded cells and region of interest for individual cells for intensity measurements (white line). (**F**) Representative calcium traces of hALN spontaneous activity before stimulation with 10 nM NPFF from individual hALN (each color represents five single cells randomly selected from 3 separate experiments with 22–56 cells per observation field in each recording). Data are presented as fluorescence intensity in arbitrary units.

## Data Availability

Single-cell RNA-seq gene expression matrices were downloaded from the Gene Expression Omnibus datasets GSE93374 (GEO Accession viewer (Gene Expression Omnibus datasets GSE93374nih.gov)) [38], GSE125065 (GEO Accession viewer (GEO Accession viewer (nih.gov)nih.gov)) [40] and GSE74672 (GEO Accession viewer (GEO Accession viewer (nih.gov)nih.gov)) [39] and from Mendeley Data dataset (Multimodal Analysis of Cell Types in a Hypothalamic Node Controlling Social Behavior in Mice—CaltechTHESIS) [41].

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
