# Peer review of "NPFF Decreases Activity of Human Arcuate NPY Neurons: A Study in Embryonic-Stem-Cell-Derived Model"

_ijms, 2022, doi:10.3390/ijms23063260_

Round 1

Reviewer 1 Report

Dear Lola Torz, dear Coauthors,

thank you for the opportunity to read your manuscript on the establishment of a human ARC cell culture and the effect of NPFF on NPY neurons.

I have only one major and few minor comments (see below). If you can respond to these, I do think the manuscript is suitable for publication.

Major comment:

Figure 2C: In the picture, you show of cells stained with DAPI and NPFFR2 I only count 5-6 cell nuclei, of which two are on the very right bottom edge of the photograph. Only two of the 5-6 cells how a significant anti-NPFFR2-staining. In the results section (line 203) you state, “NPFFR2 was present in the majority of hALNs.” To support that statement you should show a photo with more cells an in which more than half of them do stain for NPFFR2.

Minor comments:

Line 191: With “SST” do you mean Sst = somatostatin, as explained in line 107? The gene is sometimes in capital letters (line 191, table S4 and S5), sometimes not (line 107, table S2).

line 208: I am confused... with “eSC derived NPY neurons” do you mean your hALN? While reading your manuscript I spontaneously thought, ”Why not use those cells then, if they are better?” I would include “(in this case, eSC derived NPY neurons as in the differentiated hALN cell cultures we used/established)”.

legend to Figure 3, line 231 and 232: if each color represents a single cell, what is the meaning of 22-56 cells per observation field? do you mean “n=5” or what do you mean by “n=s5”?

lines 262-264: This statement is lacking a literature reference.

discussion – lines 332 ff: If feasible, you could include the possible problem, that NPFF could increase the sympathetic drive in obesity, that is already stimulated by the hyperleptinemia.

Author Response

Response to Reviewer 1 comments

Major comment

Point 1. Dear Lola Torz, dear Coauthors, thank you for the opportunity to read your manuscript on the establishment of a human ARC cell culture and the effect of NPFF on NPY neurons.

I have only one major and few minor comments (see below). If you can respond to these, I do think the manuscript is suitable for publication.

Figure 2C: In the picture, you show of cells stained with DAPI and NPFFR2 I only count 5-6 cell nuclei, of which two are on the very right bottom edge of the photograph. Only two of the 5-6 cells show a significant anti-NPFFR2-staining. In the results section (line 203) you state, “NPFFR2 was present in the majority of hALNs.” To support that statement you should show a photo with more cells an in which more than half of them do stain for NPFFR2.

Response 1: We thank Reviewer 1 for the positive feedback and suggestions as well as for the opportunity to improve the manuscript. The observation made by Reviewer 1 is accurate and, to be consistent with the data, a new image is provided (Figure 2C). The discrepancy in the previous version is explained by the fact that during the 3 weeks of maturation, a number of neurons become apoptotic and while the cells lose soma and extensions, the pycnotic nucleus still stains very brightly with DAPI (like in cells in the lower right corner). These cells can be seen in all images and are excluded from the cell counts.

Minor comments

Point 2. Line 191: With “SST” do you mean Sst = somatostatin, as explained in line 107? The gene is sometimes in capital letters (line 191, table S4 and S5), sometimes not (line 107, table S2).

Response 2: Sst refers to mouse somatostatin and SST refers to human somatostatin according to the nomenclature. All gene names are now in italics throughout the manuscript as well as in the supplementary tables 2, 3, 4, and 5)

Point 3. line 208: I am confused... with “eSC derived NPY neurons” do you mean your hALN? While reading your manuscript I spontaneously thought, ”Why not use those cells then, if they are better?” I would include “(in this case, eSC derived NPY neurons as in the differentiated hALN cell cultures we used/established)”.

Response 3: Thank you for this comment. It was indeed not clear that we refer to our own hALN and the sentence was modified as follow: ‘in this case, NPY neurons in the matured hALN population’ (line 206-207)

Point 4. legend to Figure 3, line 231 and 232: if each color represents a single cell, what is the meaning of 22-56 cells per observation field? do you mean “n=5” or what do you mean by “n=s5”?

Response 4: The ‘s’ in ‘n = s5’ has now been removed as it was a typo. Here we show calcium dynamics traces from 5 randomly selected cells from a microscopy field containing 22-56 cells We have now changed the sentence to ‘(each color represents five single cells randomly selected from 3 separate experiments with 22 – 56 cells per observation field in each recording,) ‘(line 229-231).

Point 5. lines 262-264: This statement is lacking a literature reference.

Response 5: The literature references to support the statement were added to the manuscript.

Point 6. lines 332: If feasible, you could include the possible problem, that NPFF could increase the sympathetic drive in obesity, that is already stimulated by the hyperleptinemia.

Response 6: We thank Reviewer 1 for this suggestion and the following sentence was included in the text: ‘In obesity, sympathetic tone is increased due to hyperleptinaemia and activation of HPA axis may further elevate it resulting in increased blood pressure and tachycardia.’ (Line 342)

In addition, as the editor suggested, we added the source of neuropeptide FF (NPFF) (Line 458 and 491) and add information about authors employment at Novo Nordisk to reflect potential conflict of interest in the Conflicts of Interest section (line 544-546)

Reviewer 2 Report

The manuscript by Torz et al is an interesting study aiming to generate new in vitro tools for the study of potential anti-obesity targets in humans. However, some statement throughout the manuscript are too bold and overstated from my point of view.

Saying that this human cell-derived model is a hypothalamic ARC-like neurons is too bold. I suggest to change the name to an orexigenic hypothalamic like neurons or similar. The expression pattern observed in figure 2B is not supportive enough. What about AgRP? What about POMC? The levels are insignificant. It is known, that a big proportion of NPY neurons co-express AgRP as well…in this specific case, this cell line is a representation of the NPY neurons that only express NPY. What about the neurons that co-express both? And it is very important, at least if this line should work as a tool for anti-obesity drugs, to have the counterpart representation (POMC anorexigenic neurons) to really test the effects of a potential drug. Moreover, NPY, TH, and others are not exclusively express in the ARC of the hypothalamus, making the lack of AgRP and POMC a concern.

Figure 1D seems to be incomplete to my advice. Authors need to do an in situ hybridization staining of NPFFR2 together with NPY, and maybe NPFFR2 together with POMC as a negative control, or in order to show the more expression of NPFFR2 in NPY neurons.

Figure 1E. Why the NPFFR2 staining looks mainly nuclear? Do the authors have any plausible explanation? In figure 1D (mouse) the expression pattern is a mix between nuclear and cytosolic.

I suggest to add the result of the stimulation with 50 mM KCL induction of cytoplasmic calcium. In the text (line 256) is written data not shown. The authors have enough space in the supplementary data. I recommend to add this piece of information for the sake of transparency.

The calcium experiments are very elegant, however, are these neurons really functional? What about the neuropeptide levels after the treatment of forskolin and NPFF?

Author Response

Response to Reviewer 2 comments

 Point 1. The manuscript by Torz et al is an interesting study aiming to generate new in vitro tools for the study of potential anti-obesity targets in humans. However, some statement throughout the manuscript are too bold and overstated from my point of view. Saying that this human cell-derived model is a hypothalamic ARC-like neurons is too bold. I suggest to change the name to an orexigenic hypothalamic like neurons or similar. The expression pattern observed in figure 2B is not supportive enough. What about AgRP? What about POMC? The levels are insignificant. It is known, that a big proportion of NPY neurons co-express AgRP as well…in this specific case, this cell line is a representation of the NPY neurons that only express NPY. What about the neurons that co-express both?

Response 1: We thank Reviewer 2 for the insight full suggestion and constructive points raised.

A similar protocol was used by several groups to generate hypothalamic cultures referred to as “arcuate-like” (Rajamani et al. (2018) and Wang et al. (2015) (also referenced 34 and 26, respectively, in the manuscript). Our assessment of cell type to a large degree matches with the published data, as we also find several specific markers of arcuate neurons: Agrp (albeit at much lower levels than NPY) and POMC by immunostaining and NPFFR2. In addition, low gene expression levels go together with the presence of clear single puncta of Agrp immunoreactivity. Low mRNA detection for Agrp and POMC reflect the fact that gene expression data represent a snapshot in cell’s time course, and mRNA levels do not always translate 1:1 to protein expression.

Although we did not detect an increase in POMC expression in matured hALNs compared to NSC, we observed 0.1% of POMC-immunoreactive neurons similar to what Merkel et al. (2015) reported (less than 1 per 1000 nuclei, Figure 5G, Merkle et al. (2015)).

Thus, we believe that the presence of arcuate-specific markers allows for more extensive studies in this cell platform to study responses from these cell types as a model for both orexigenic and anorexigenic signaling in the arcuate nucleus. We hope that Reviewer 2 will find this point sufficiently substantiated, and we can keep the existing abbreviation. 

Point 2. It is very important, at least if this line should work as a tool for anti-obesity drugs, to have the counterpart representation (POMC anorexigenic neurons) to really test the effects of a potential drug. Moreover, NPY, TH, and others are not exclusively express in the ARC of the hypothalamus, making the lack of AgRP and POMC a concern

Response 2: The overall statement expressed by Reviewer 2 is entirely correct, and we are planning to develop methods for identifying POMC neurons in our cultures and record their responses. In this study, for performing pharmacology assays and ligand-receptor interactions, a homogenous system would have a clear advantage over highly heterogenous population. We mention this point in lines 206-209 in the manuscript. In addition, studying responses from different interconnected neuronal types would require introduction of a cell reporter into the system. So far, we were not able to obtain or generate a human hypothalamic line with a specific neuron type reporter system, but we are working towards this goal. 

Point 3. Figure 1D seems to be incomplete to my advice. Authors need to do an in situ hybridization staining of NPFFR2 together with NPY, and maybe NPFFR2 together with POMC as a negative control, or in order to show the more expression of NPFFR2 in NPY neurons.

Response 3: The best option for the mouse tissue here would be a duplex ISH to Npffr2/Pomc and Npffr2/Npy, however this would require ordering new probes and with the current health situation the probes delivery is highly delayed which make it impossible for us to include this experiment within the revision time. In our opinion, the gene expression profile in mouse NPFFR2 neurons is already presented in the paper, but the characterization of human NPFFR2 neurons has not yet been done and therefore NPFFR2/POMC staining, as Reviewer 2 suggested, would improve the study. We have performed this co-detection with ISH and found that the expression of NPFFR2 in human POMC neurons is either not detected or very low expressed. The image is now added to Figure 1E and in the manuscript (line147).

Point 4. Figure 1E. Why the NPFFR2 staining looks mainly nuclear? Do the authors have any plausible explanation? In figure 1D (mouse) the expression pattern is a mix between nuclear and cytosolic.

Response 4: This discrepancy may be due to more compact versus diffuse ER anatomy between species, so in the examples of more “nuclear” staining, the signal (ER + mRNA to Npffr2) are located in a tighter proximity to the nucleus. In the past, we observed such differences across various cell types and between species. In addition, the NPFFR2 ISH expression was detected at lower levels in human ARC and we only chose neurons which had a clear expression (several dots) around the nucleus, to be sure that they belong to the same cell. The scattered NPFFR2 expression is also present in the cytoplasm and in axons, but it is more difficult to accurately determine which neuron this signal belongs in thin FFPE sections from human brain, since the nuclei are rather large and the expression levels from receptors are typically much lower than for the neuropeptides.

Point 5. I suggest to add the result of the stimulation with 50 mM KCL induction of cytoplasmic calcium. In the text (line 256) is written data not shown. The authors have enough space in the supplementary data. I recommend to add this piece of information for the sake of transparency.

Response 5: We have now added the figure representing the calcium response of the hALN cells to 50 mM KCL into a new supplemental figure 2 as mentioned line 255 and figure legend was added line 511-512.

Point 6. The calcium experiments are very elegant, however, are these neurons really functional? What about the neuropeptide levels after the treatment of forskolin and NPFF?

Response 6: This is an excellent suggestion. However, because changes in the calcium oscillation pattern drive neuromediator release, it would be very difficult to follow the temporal resolution (less than one minute medium sampling) for several neuropeptides and GABA. While such experiment is not possible to perform within the revision time (also because of long neuron maturation time), we will work to adapt this neuronal platform for neuropeptide release assays with high temporal resolution. On the other hand, the relation of neuronal activity and intracellular calcium dynamics is well established and, in our opinion, allows for interpretation of NPFFR2 action.  

In addition, as the editor suggested, we added the source of neuropeptide FF (NPFF) (Line 458 and 491) and add information about authors employment at Novo Nordisk to reflect potential conflict of interest in the Conflicts of Interest section (line 544-546)

Round 2
